# Prediction of Seedling Oilseed Rape Crop Phenotype by Drone-Derived Multimodal Data

**Yang Yang [1], Xinbei Wei [1], Jiang Wang [1], Guangsheng Zhou [2,*], Jian Wang [2], Zitong Jiang [2], Jie Zhao [2] and Yilin Ren [1]**

[1] School of Engineering, Huazhong Agricultural University, Wuhan 430070, China; yangyang@mail.hzau.edu.cn (Y.Y.); xinbei@webmail.hzau.edu.cn (X.W.); wangjiang201@webmail.hzau.edu.cn (J.W.); renyilin@mail.hzau.edu.cn (Y.R.)

[2] College of Plant Science and Technology, Huazhong Agricultural University, Wuhan 430070, China; wangjmail.hzau.edu.cn@webmail.hzau.edu.cn (J.W.); jiangzt@webmail.hzau.edu.cn (Z.J.); zhaojie@mail.hzau.edu.cn (J.Z.)

* Correspondence: zhougs@mail.hzau.edu.cn

**Abstract:** In recent years, unmanned aerial vehicle (UAV) remote sensing systems have advanced rapidly, enabling the effective assessment of crop growth through the processing and integration of multimodal data from diverse sensors mounted on UAVs. UAV-derived multimodal data encompass both multi-source remote sensing data and multi-source non-remote sensing data. This study employs Image Guided Filtering Fusion (GFF) to obtain high-resolution multispectral images (HR-MSs) and selects three vegetation indices (VIs) based on correlation analysis and feature reduction in HR-MS for multi-source sensing data. As a supplement to remote sensing data, multi-source non-remote sensing data incorporate two meteorological conditions: temperature and precipitation. This research aims to establish remote sensing quantitative monitoring models for four crucial growth-physiological indicators during rapeseed (*Brassica napus* L.) seedling stages, namely, leaf area index (LAI), above ground biomass (AGB), leaf nitrogen content (LNC), and chlorophyll content (SPAD). To validate the monitoring effectiveness of multimodal data, the study constructs four model frameworks based on multimodal data input and employs Support Vector Regression (SVR), Partial Least Squares (PLS), Backpropagation Neural Network (BPNN), and Nonlinear Model Regression (NMR) machine learning models to create winter rapeseed quantitative monitoring models. The findings reveal that the model framework, which integrates multi-source remote sensing data and non-remote sensing data, exhibits the highest average precision ($R^2 = 0.7454$), which is 28%, 14.6%, and 3.7% higher than that of the other three model frameworks, enhancing the model's robustness by incorporating meteorological data. Furthermore, SVR consistently performs well across various multimodal model frameworks, effectively evaluating the vigor of rapeseed seedlings and providing a valuable reference for rapid, non-destructive monitoring of winter rapeseed.

**Keywords:** machine learning; multi-source data fusion; nitrogen; oilseed rape; unmanned aerial vehicle (UAV)

## 1. Introduction

Field crop phenotypic information refers to the physical, physiological, and biochemical characteristics of crop growth and development [1], such as the leaf area index (LAI), above-ground biomass (AGB), leaf nitrogen content (LNC), and chlorophyll content, which are influenced by internal and external environmental factors [2–4]. These growth physiological parameters are important plant indicators for dynamic monitoring of vegetation growth and reflect the growth of crops [5]. The seedling stage has a decisive influence on growth, development, and yield formation, and growth monitoring of winter rape seedlings plays an important role in decision making for field production and crop regulation [6]. Traditional measurement methods rely on manual collection in the field,

which is not only time-consuming and laborious, but may also cause some damage to the plant, while lacking real-time and spatial distribution accuracy [7]. Therefore, rapid, accurate, and nondestructive measurement of plant biomass is of great value in all aspects of precision agriculture.

To address this problem, research on the use of unmanned aerial vehicle (UAV) remote sensing to estimate crop growth and physiological parameters has been emerging. Remote sensing, as a cutting-edge information technology for terrain observation, can be used to quickly and accurately obtain real-time information on crop growth and physiology over large areas, and has been widely used in agriculture in recent years [8–10]. Multispectral images have the advantages of high spatial resolution and ease of operation [11]. Monitoring of field crops can be achieved quickly and nondestructively using the broadband extracted by UAVs carrying multispectral cameras in combination with existing spectral indices [12,13]. Many studies have shown that, by combining UAV multispectral images with high spatial resolution vegetation indices (VIs) and using existing mature machine learning algorithms, reliable models can be built to effectively and nondestructively monitor plant growth [14–16]. The high-spectral, multispectral (MS), and visible images (RGB) captured through UAV-based remote sensing exhibit distinct characteristics. Integrating these images with machine learning algorithms enables robust monitoring of crop growth. Nevertheless, the simultaneous leveraging of the advantages offered by multiple sensors remains underexplored.

With the advancements in remote sensing and agricultural technology, it is now possible to acquire a variety of remote sensing image data and non-remote sensing data, such as meteorological and soil data, from multiple sensors and time periods within the same geographical area. These datasets collectively form the multimodal data within the region [17,18]. Multimodal data refers to the fusion of diverse data sources, synthesizing the image information of multiple imaging sensors for the same target. Effective integration of the complementary information from different data sources mitigates the limitations of single-source data, including incomplete interpretation, uncertainty, and errors associated with monitoring the target, thereby enhancing the efficiency and depth of utilizing multi-source data [19,20]. Multimodal data can be categorized into two parts: (1) fusion between multi-source remote sensing data; (2) fusion between multi-source remote sensing data and multi-source non-remote sensing data.

In the field of multi-source remote sensing data fusion, fusion of multi-source data can make up for the limitations of a single image and increase the quality of experience (QoE), while enhancing the quality of remote sensing images. In addition, high-resolution remote sensing images have a positive impact on the accuracy of the subsequently constructed models [21]. There is a growing interest in using multi-source data to estimate crop growth in the field because fusing multi-source data can compensate for the limitations of a single image, and many studies have established high-precision nondestructive estimation models based on UAV remote sensing [22–24]. This research has shown that multimodal features are more advantageous than single-modal features, and can improve the feasibility and accuracy of the model in many ways, such as by fusing audio–visual information to improve some unimodal visual analysis systems [25–27]. However, using all VIs and texture features (Texs) obtained from different remote sensing sensors only as inputs to the monitoring models may lead to data redundancy and problems such as multicollinearity, which instead reduce the robustness of the monitoring models [28]. In addition, multi-source non-remote sensing data are underutilized in these models. Therefore, many scholars have proposed methods with deeper levels of fusion, such as remote image fusion methods in the multi-scale morphological gradient (MSMG) structural domain [29], and the hybrid fusion method of IR and visual images combining discrete smooth wavelet transform (DSWT), discrete cosine transform (DCT), and local spatial frequency (LSF) [30], to fuse multi-source remote sensing data using image fusion algorithms to enhance the spatial resolution of remote sensing images while maintaining the original information of the spectrum.

The fusion between multi-source non-remote sensing data and multi-source remote sensing data mainly involves the participation of non-remote sensing information, such as meteorological data, soil data, and geographic information, as auxiliary variables in remote sensing monitoring and classification applications [31]. Existing studies have demonstrated that local changes in the crop canopy as a response to environmental and field management changes affect crop yield, suggesting that non-remote sensing information as a feature may have a powerful role in crop yield prediction by incorporating non-remote sensing data in model training, and that visible or multispectral images can yield better prediction results [32–34]. However, there is a dearth of research on image fusion algorithms combined with multi-source non-remote sensing features to estimate the phenotypic information of oil-seed rape, which poses challenges in ensuring the stability of models in predicting crop growth information in different fields.

Based on this situation, an oil-seed rape test field in Shashi District, Jingzhou City, Hubei Province, was used as the study area in this study. Employing an enhanced image fusion algorithm, the study merges UAV visible light images with multispectral images. Subsequently, four widely used machine learning prediction methods—PLSR, NLR, SVR, and BP-NN—are harnessed to effectively integrate diverse non-remote sensing data sources for monitoring the growth and physiological parameters of rapeseed in the field. The main objectives were to (1) apply four regression methods (PLSR, NLR, SVR, and BP-NN) to establish a model for monitoring growth and physiological parameters of field rapeseed; (2) compare the performance of estimating physiological parameters of oil-seed rape growth using traditional VIs, Texs, and four input model frameworks based on multimodal data composition; and (3) compare and analyze the estimation results of the models to determine the best model for each growth index.

## 2. Materials and Methods

### 2.1. Field Experiment and Biomass Sampling

An experiment was conducted at Jingzhou Agricultural Science Academy, Shashi District, Jingzhou City, Hubei Province (112°20′35″E, 30°14′17″N). The test field covered an area of about 3800 m$^2$, as shown in Figure 1, which is a schematic diagram of the field test area and the UAV remote sensing photography. The oil-seed rape cultivar was mainly Huayouza 50, jointly developed by Huazhong Agricultural University and Wuhan Liannong Seed Technology Co., Ltd. (Wuhan, China) with registration number GPD Oil-seed rape (2017) 420204. This trial was conducted from September 2022 to March 2023, depending on the developmental progress of winter oil-seed rape. A single-factor experiment was set up as follows: three N application levels: 8 kg/667 m$^2$ (N8), 12 kg/667 m$^2$ (N12), 16 kg/667 m$^2$ (N16); three density treatments: 10,000 plants/667 m$^2$ (D1), 30,000 plants/667 m$^2$ (D3), 50,000 plants/667 m$^2$ (D5); three sowing periods: September 25 (S925), October 10 (S1010), and October 25 (S1025). The distribution of the rape trial area and treatments are shown in Figure 2. The trial was set up with multiple replications; each plot area was 2 m × 2 m, row spacing was 0.5 m, the whole trial field was shaped like a trapezoid, and multiple protection rows were set up with a total of 546 plots. Except for the above treatment differences, other management measures were the same as the local high-yielding cultivation measures.

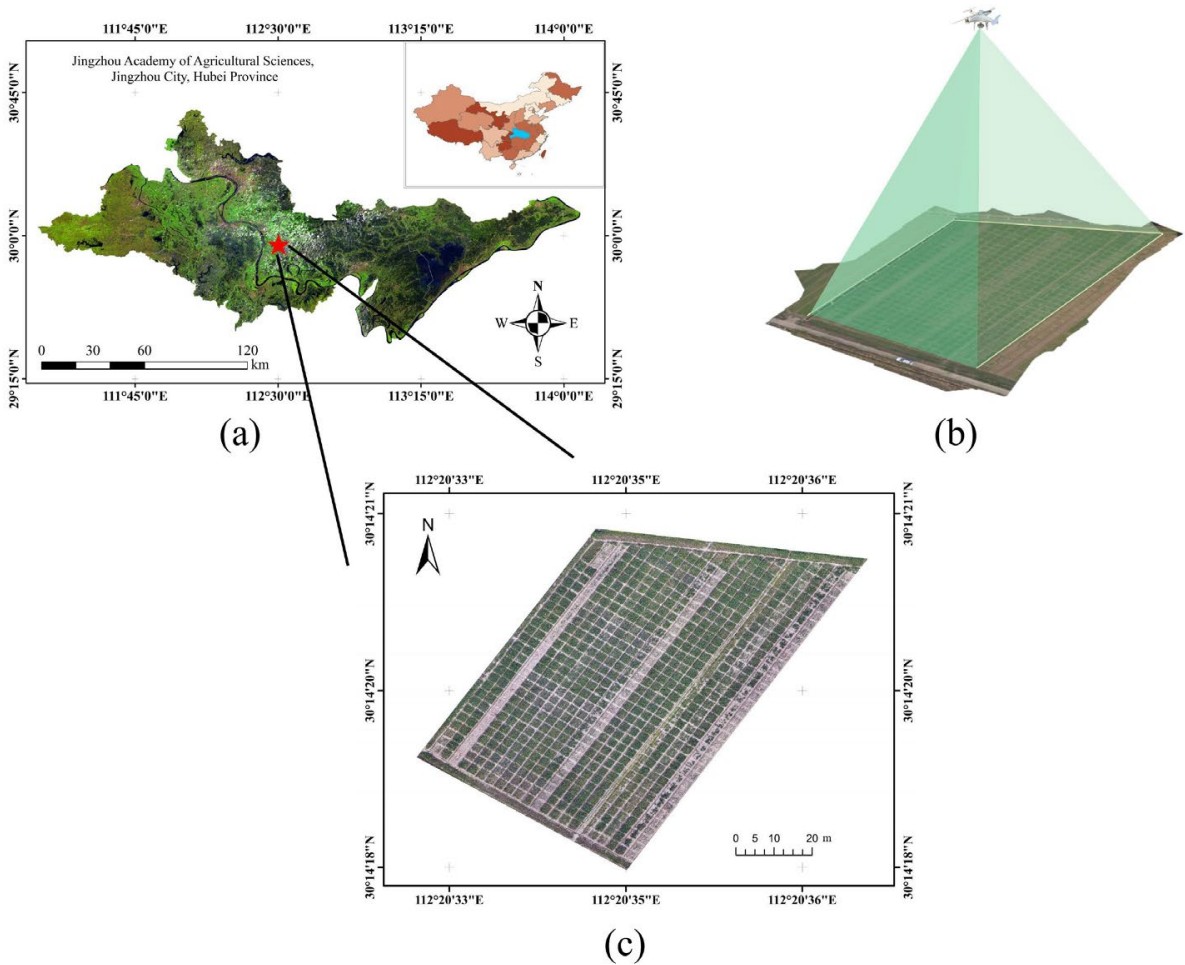

**Figure 1.** Field diagram of the test site and UAV: (**a**) location of the research field in Hubei Province; (**b**) aerial view of UAV; (**c**) location of sampling plots.

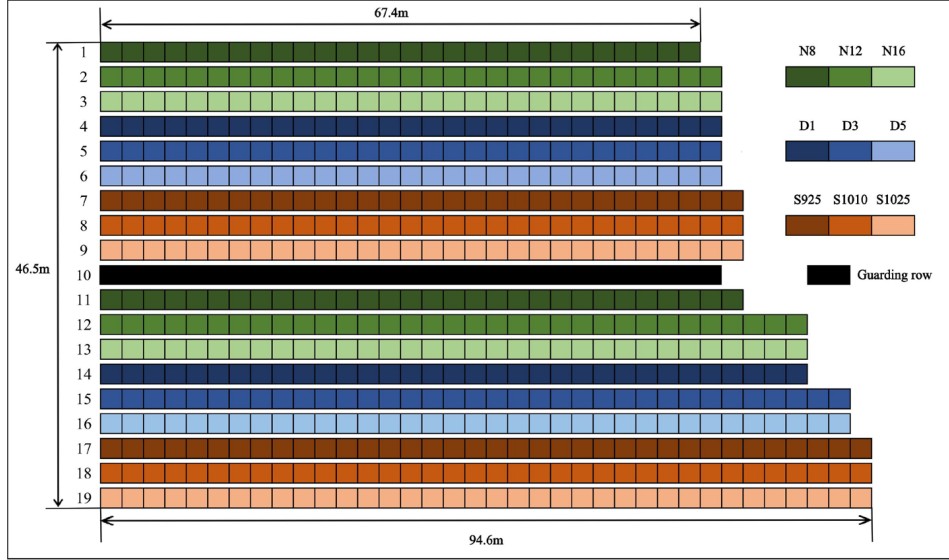

**Figure 2.** Distribution of rape experimental area and treatment. N8, N12, N16 represent three N application levels; D1, D3, D5 represent three density treatments; S925, S1010, S1025 represent three sowing periods.

### 2.2. Collection and Processing of Rapeseed Phenotype Information

During the experiment, seedling oil-seed rape data, including UAV-based multispectral imaging data, LAI, SPAD, LNC, and AGB, were collected from November 2022 to March 2023, using an AccuPAR LP80 racing Radiation and Architecture of Canopies meter, a SPAD-502 chlorophyll meter, an NKY-6120 Nitrogen Analyzer, and electronic scales (Figure 3), in four consecutive collections during the critical fertility period of winter oil-seed rape in 21 November 2022, 8 December 2022, 10 January 2023, and 30 January 2023.

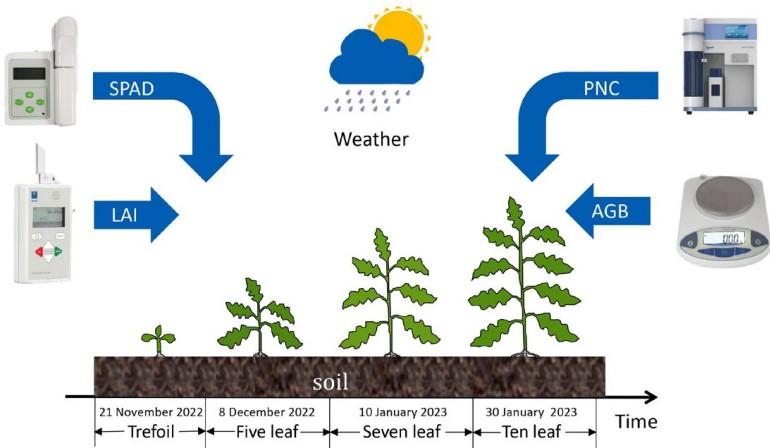

**Figure 3.** LAI, SPAD, PNC, and AGB and weather data collection at the rapeseed seedling stage.

### 2.3. UAV Systems and Flight Missions

The UAV used a DJI Mavic 3M (Figure 4a) in this test to simultaneously collect RGB and multispectral imagery. The total weight of the UAV was 1.05 kg, the maximum pitch angle was $35°$, the maximum horizontal flight speed was 21 m/s, and the flight endurance was about 43 min. Image acquisition was performed using the visible light camera and multispectral camera (Figure 4b) equipped with the UAV, and the details of the sensors are shown in Table 1. The choice of the wavelength is very important for the calculation of the VIs but, due to equipment limitations, the bandwidth and wavelength shift issues are not taken into account in the calculations, and multiple central band data from multispectral cameras are directly used [35]. The UAV was also equipped with a multispectral light intensity sensor on top of the UAV, which can monitor the incident light intensity in real time and compensate for multispectral imaging. The DJI RC Enterprise (Figure 4a) was used to automatically generate the flight routes, and Figure 4c shows the two-dimensional (2D) routes based on satellite map data as well as the three-dimensional (3D) routes based on the UAV elevation data.

**Table 1.** Sensor parameters of the UAV.

| Sensor Category | Spectral Area (μm) | Resolution | Field of View (H° × V°) |
|---|---|---|---|
| Visible light | N/A | 1600 × 1200 | 56° × 84° |
| Multispectral | Green: 0.560; Red: 0.650; Red edge: 0.730; NIR: 0.860 | 800 × 600 | 47.2° × 73.9° |

Prior to the UAV mission, the heading/bypass overlap was set to 75% and 70%, respectively, and the maximum flight speed was 5 m/s, and the shooting time was selected between 10 a.m. and 2 p.m. when the weather was clear and there was direct sunlight. Four flights were conducted to simultaneously acquire UAV imagery at a flight altitude of 40 m and collect phenotypic data of oil-seed rape. Table 2 shows the acquisition of remote sensing data and field trial data.

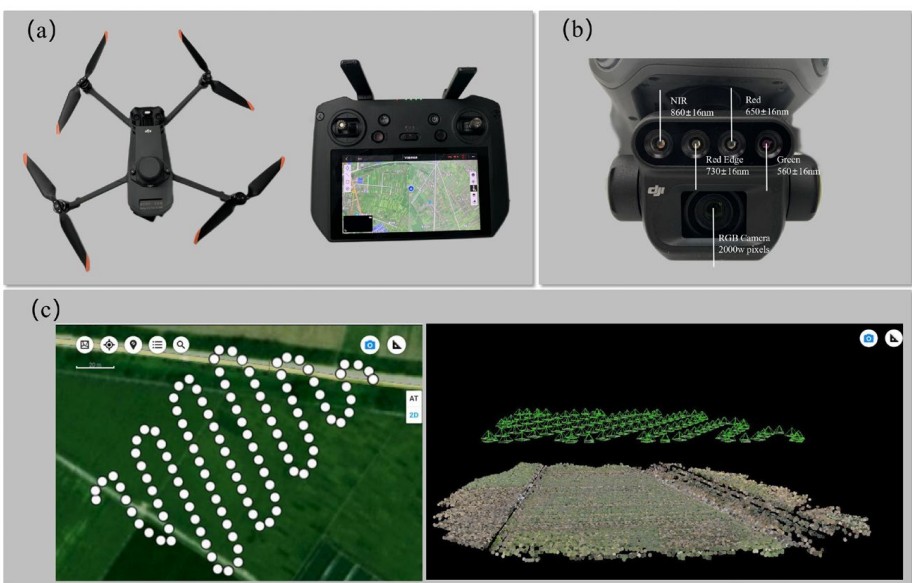

**Figure 4.** UAV remote sensing device and flight control system: (**a**) DJI RC Enterprise, UAV, and flight controller; (**b**) multispectral and visible light sensor of UAV; (**c**) route planning (2D and 3D schematic).

**Table 2.** Acquisition of remote sensing data.

| Test Time | Date of Remote Sensing Image Acquisition | Precise Time | Height (m) | Heading/Sideways Overlap |
|---|---|---|---|---|
| 19 November 2022–21 November 2022 | 21 November 2022 | 11:30–12:00 | 40 | 75/70% |
| 8 December 2022–10 December 2022 | 8 December 2022 | 12:00–12:30 | 40 | 75/70% |
| 9 January 2023–11 January 2023 | 10 January 2023 | 12:00–12:30 | 40 | 75/70% |
| 29 January 2023–31 January 2023 | 30 January 2023 | 13:30–14:00 | 40 | 75/70% |

Note: The acquisition of remote sensing data and field trial data was completed simultaneously during the trial time.

### 2.4. Image Processing and Feature Extraction

#### 2.4.1. Image Processing

The detector relative spectral response (RSR) shift effect affects the uniformity of the collected radiation signals. In this study, the camera calibration function in Metashape software was used to align and correct the UAV images, and operations such as image alignment, dense point cloud creation, grid generation, texture generation, and elevation image generation on the UAV images were carried out sequentially to obtain a high-resolution digital orthophoto map (DOM) [36]. Hazy scenes reduce the feasibility of image analysis; however, due to the suitable environment and proper conditions of the shooting process, the images collected by the UAV in this study can be regarded as fog-free images, and a de-fogging process was not needed [37]. The RGB images and the four-band MS images are stitched together to cover the whole test area, where MS_G, MS_R, MS_B, and MS_NIR represent the corresponding single-band spectral images. The stitched RGB images were processed using MATLAB 2020b, and the multispectral band images were independent (i.e., MS_G, MS_R, etc.). The band fusion of the MS image set was implemented using ArcGIS software, and the fusion mode of standard pseudo-color was selected to assign the three bands of NIR MS_NIR, red MS_R, and green MS_G to red, green, and blue colors, respectively, and the obtained images of vegetation or crops in red color. The bit depth of MS images after band fusion is 32 bits, and the remote sensing image is rendered to a depth of 8 bits, so that the rendered image after band fusion is processed in ArcGIS and can be read and processed by software such as MATLAB and Python. Finally, the imported band-fused standard dummy color large-field images are cropped using ENVI software to obtain images of individual fields.

### 2.4.2. VIs

In remote sensing applications, vegetation indices have been widely used to qualitatively and quantitatively evaluate vegetation cover and its growth vigor [38]. The vegetation indices used in this study are shown in Table 3.

**Table 3.** Calculation of spectral index.

| Spectral Index | Abbreviations | Calculation Formula | Source |
|---|---|---|---|
| Normalized vegetation index | NDVI | NDVI = (NIR − R)/(NIR + R) | [39] |
| Nitrogen reflection index | NRI | NRI = (G − R)/(G + R) | [40] |
| Greenness vegetation index | GNDVI | GNDVI = (NIR − G)/(NIR + G) | [41,42] |
| Ratio vegetation index | RVI | RVI = NIR/R | [43] |
| Non-linear vegetation index | NLI | (NIR × NIR − R)/(R × R − NIR) | [44] |
| Modified simple ratio index | MSR | (NIR/R − 1)/[(NIR/R + 1)^(1/2)] | [45] |

### 2.4.3. GCFs

In order to improve the accuracy of the model, many scholars further extract texture features from each multispectral-based band to construct the model. In this study, a texture parameter based on gradual change features (GCFs) calculated from NDVI was chosen to construct a new texture index by obtaining the structural distribution characteristics of the spectral indices. NDVI, as one of the most commonly used spectral indices in agricultural remote sensing monitoring applications, is highly sensitive to changes in crop growth and physiological parameter indices, and can well distinguish between crop groups with different measures of growth potential [46]. In a 2 m × 2 m plot, a 1 m × 1 m area in the center was selected, and the grayscale images of NDVI were classified into five categories according to the size of the image elements using the K-means clustering algorithm: NDVI minimum area (A), NDVI small area (B), NDVI medium area (C), NDVI large area (D), and NDVI maximum area (E); the number of image elements occupied by each category was counted to characterize the area occupied by each category. The flow chart of this process is shown in Figure 5.

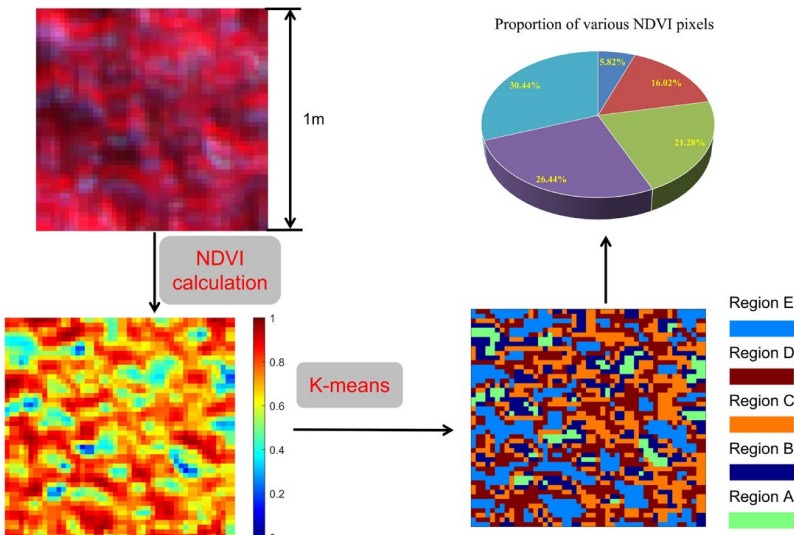

**Figure 5.** Acquisition process of a texture parameter based on GCFs.

The NDVI values of the five types of areas were averaged as $V_A$, $V_B$, $V_C$, $V_D$, and $V_E$, and the image elements were recorded as the area of the five areas as $S_A$, $S_B$, $S_C$, $S_D$, and $S_E$. Generally, the areas with NDVI values less than 0.2 can be classified as non-vegetation covered areas. The $V_A$ and $S_A$ corresponding to this part can be discarded to improve the accuracy of the gradient feature data. Four gradient feature indicators were designed for

four types of NDVI and area share, which are the vegetation index coefficient of variation ($V_{cv}$), area coefficient of variation ($S_{cv}$), vegetation compactness ($R_c$), and vegetation density coefficient of variation ($R_a$), as expressed in Equations (1)–(6):

$$V_m = \frac{V_B + V_C + V_D + V_E}{4} \tag{1}$$

$$S_m = \frac{S_B + S_C + S_D + S_E}{4} \tag{2}$$

$$V_{cv} = \sqrt{\frac{(V_B - V_m)^2 + (V_C - V_m)^2 + (V_D - V_m)^2 + (V_E - V_m)^2}{4}} \tag{3}$$

$$S_{cv} = \sqrt{\frac{(S_B - S_m)^2 + (S_C - S_m)^2 + (S_D - S_m)^2 + (S_E - S_m)^2}{4}} \tag{4}$$

$$R_c = \frac{S_B}{4 \times S_m} \tag{5}$$

$$R_a = \frac{V_E - V_B}{4 \times V_m} \tag{6}$$

### *2.5. Non-Remote Sensing Auxiliary Data*

Non-remote sensing data can be used as a supplement to remote sensing data and help to improve the scientific nature of the study. Daily maximum temperature ($T_{max}$), minimum temperature ($T_{min}$), and average rainfall ($A_{ar}$) were selected as non-remote sensing data, and the data were obtained from the National Weather Science Data Center (http://data.cma.cn, accessed on 16 March 2023). The experiment collected meteorological data of the area where Daejeon is located, from 25 September 2022 to 30 January 2023. The daily average rainfall can be calculated by monthly average rainfall (monthly average rainfall divided by the number of days in the month), and the daily average rainfall was summed according to the time interval of remote sensing monitoring in this study as the rainfall data for the monitoring model.

The maximum temperature, $T_{max}$, and minimum temperature, $T_{min}$, in the non-remote sensing data are time series and dynamic, which are difficult to combine effectively with the remote sensing data collected in multiple time intervals. Therefore, this study introduced the effective cumulative temperature, $A_e$, to convert the time series continuous data into non-temporal discrete data, which was the sum of the effective temperature of the crop at a certain reproductive period and, numerically, the sum of the difference between the average temperature at a certain time period and the biological zero of the crops, as in Equation (7) [47]:

$$A_e = \sum_{i=1}^{n} (T_i - B) \tag{7}$$

where $T_i$ is the average temperature during the $i$th period; $B$ is the biological zero, which is the minimum temperature required to meet the crop's continued growth and development, and is related to the crop's species and development time. The biological zero of the oil-seed rape species in this study is generally between 4 °C and 5 °C, and here $B = 4$ °C was chosen.

## 3. Multimodal Data Fusion

Multi-source data fusion can be divided into two parts: (1) the fusion between multi-source remote sensing data; (2) the fusion between multi-source remote sensing data and multi-source non-remote sensing data. The former fully fuses UAV multispectral images and visible images using image fusion algorithms to realize the enhancement of image

information, and the latter realizes the organic combination of multiple variables through machine learning algorithms.

### 3.1. Image Fusion

In this study, an edge-preserving filtering algorithm, guided filtering (GF), which was developed by Li et al. [48], was chosen. The algorithm is based on a local linear model to guide the information of the digital image to calculate the filtering output, which can be used in applications such as upsampling, local cropping, color space conversion, and multi-scale decomposition in image processing. In the multi-scale transformation session, GF takes another image as the guide, and this guide image can be the input or even the same image of this decomposition layer. By analyzing the distribution of pixel neighborhoods, a linearly invariant output image is generated, consisting primarily of an approximation image and a structure image. This approach effectively preserves the structural characteristics of the source image, facilitating efficient multi-scale decomposition.

The study performs multi-scale decomposition using GF, and the decomposed multi-scale representation can be reconstructed as a source image using Inspiratory Muscle Strength Training (IMST), as shown in Figure 6:

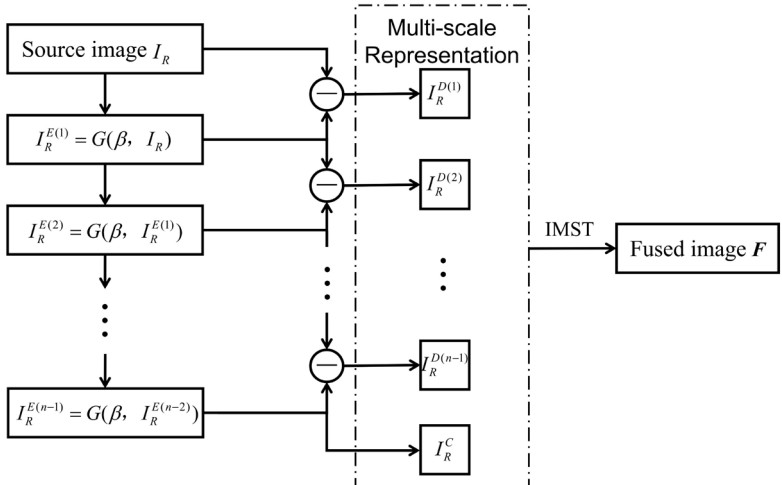

**Figure 6.** Multi-scale decomposition of GF.

Suppose the source image used for multi-scale decomposition is $I_R$, and the operator performing the decomposition operation is named $G(\bullet)$, as shown in Equation (8):

$$
\begin{aligned}
&\exists \text{ operator } G \text{ , } I_{out} = G(\beta, I_{in}) \\
&\text{where } I_{out} = \arg_{I_{out}} \min \|I_{in} - I_{out}\|_2^2 + \beta\left(\|\alpha_x G_x I_{out}\|_2^2 + \|\alpha_y G_y I_{in}\|_2^2\right) \\
&\alpha_x = \left(\frac{\partial I_{in}}{\partial x}\right)^{-\alpha}, \alpha_y = \left(\frac{\partial I_{in}}{\partial y}\right)^{-\alpha}
\end{aligned} \tag{8}
$$

where $G_x$ and $G_y$ are the difference operators for horizontal and vertical directions, respectively; the parameter $\beta$ is a regularization constant to control the balance between horizontal and vertical targets; $I_{in}$ is the image used for multi-scale decomposition; and $I_{out}$ is calculated from the equations constructed by the operator $G$.

Let the images after GF decomposition be divided into two categories: an approximate image $I_R^C$ and a set of detailed images $I_R^{D(i)}$, $i = 1, 2, \ldots, n-1$; then, $I_R^C$ and $I_R^{D(i)}$ can be shown as in Equation (9):

$$
\begin{aligned}
&I_R^{E(i)} = G\left(\beta_i, I_R^{E(i-1)}\right), i = 1, 2, \ldots, n-1 \\
&I_R^{E(0)} = I_R, \quad I_R^C = I_R^{E(n-1)} \\
&I_R^{D(i)} = I_R^{E(i-1)} - I_R^{E(i)}
\end{aligned} \tag{9}
$$

where $I_R^{E(n-1)}$ is the image that has undergone $n-1$ decomposition, which has the lowest resolution; the last decomposed image is generally taken as the approximate image of GF decomposition $I_R^C$, and the difference in the approximate images between different decomposition layers is considered as the detailed image of multi-scale decomposition.

Taking a UAV remote sensing image acquired from a large field as an example, one approximate image and two detailed images obtained after three GF decompositions are shown in Figure 7. The approximate image has the same color distribution as the source image, and the detailed image shows the rape field and the leaf texture of the crop within the field, so the GF decomposition can store the color or spectral information of the source image in the approximate image $I_R^C$ and the structural and spatial information in the detailed image $I_R^{D(i)}$.

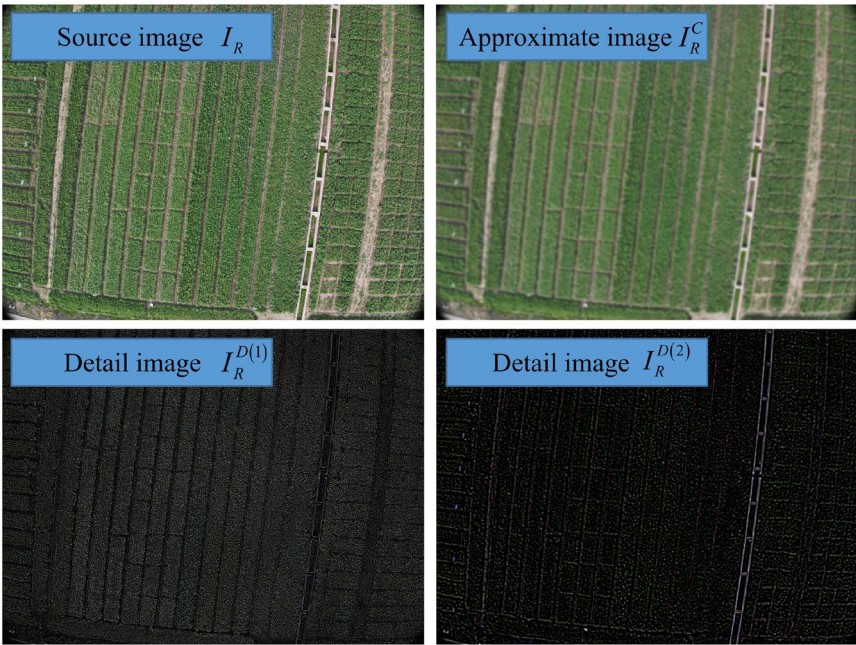

**Figure 7.** Images based on GF decomposition.

The decomposed image can construct the input source image by approximating the image and detailed image, which achieves the inverse multi-scale transformation and nearly lossless restoration of the source image. This is the theoretical basis of the multi-scale decomposition-based image fusion algorithm, and the transformation process is shown in Equation (10):

$$I_R = I_R^C + \sum_{i=1}^{n-1} I_R^{D(i)}, \quad i = 1, 2, \ldots, n-1 \tag{10}$$

In the GF image fusion algorithm, if the source image is $I_1, I_2, \ldots, I_M$ for a total of $M$ images, the approximate image $I_1^C, I_2^C, \ldots, I_M^C$ and the detailed image $I_1^{D(i)}, I_2^{D(i)}, \ldots, I_M^{D(i)}$ can be obtained after the calculation of Equations (9) and (10).

The GF decomposed image needs to be fused according to the fusion rules, given a set number of source images, where the nth image is $f$. Laplace filtering is performed on image f to obtain a high-pass image $H_n$, as shown in Equation (11):

$$H_n = h_n(f) * L(f)$$
$$h_n(f) = \frac{1}{m^2} \begin{bmatrix} 1 & 1 & \cdots & 1 \\ 1 & 1 & \cdots & 1 \\ \vdots & \vdots & \ddots & \vdots \\ 1 & 1 & \cdots & 1 \end{bmatrix} \tag{11}$$

where $L(f)$ represents the logarithmic spectrum of the image and $h_n(f)$ is an $m \times m$ matrix. Usually, $m = 3$ and the change in $m$ has a small effect on the calculation of the significance map. The local average of the absolute values of $H_n$ is used to construct the significance map $S_n$, as shown in Equation (12):

$$S_n = |H_n| * g_{rg,\delta g} \tag{12}$$

where $g$ is the low-pass filter, the size of $rg$ and $\delta g$ is $m - 1$, and the size of the low-pass filter is $[2(m - 1) + 1]^2$. The same remote sensing image of a large field of oil-seed rape exemplified in Section 3.1 is used to calculate the log spectrum and significance map of the image. Finally, the significance map extracted in the previous step is used to calculate the initial weight map, $P_n$, of the source image, which is calculated as shown in Equation (13):

$$P_n(p,q) = \begin{cases} 1 & if\, S_n(p,q) = \max\{S_1(p,q), S_2(p,q), \ldots, S_n(p,q)\} \\ 0 & otherwise \end{cases} \tag{13}$$

where $S_n(p,q)$ denotes the significant value at the pixel $(p,q)$ in the $n$th image. The weighted average of the weights of the approximation and detailed images for the best estimate is calculated as the weight maps of $P^C$ and $P^{D(i)}$, where $i$ is the number of layers of the decomposition. Then, the new fusion rule is shown in Equation (14):

$$\begin{aligned} \overline{C\prime} &= P^C \cdot I_1 + (1 - P^C)I_2 \\ \overline{D\prime} &= \sum_{i=1}^{n-1} P^{D(i)} \cdot I_1^{D(i)} + \sum_{i=1}^{n-1} (1 - P^{D(i)}) \cdot I_1^{D(i)} \\ F\prime &= \overline{C\prime} + \overline{D\prime} \end{aligned} \tag{14}$$

where $P^C(k)$ and $P^{D(i)}$ are the weight values at the $k$th pixel of the weight map, which is in the range [0, 1]. $\overline{C\prime}$ is the fused approximate image, $\overline{D\prime}$ is the fused detailed image, and $F\prime$ is the fused image.

*3.2. Machine Learning*

Machine learning algorithms possess exceptional capabilities in nonlinear regression prediction. They find increasingly widespread application in precision agriculture and remote sensing monitoring, demonstrating particularly remarkable performance in scenarios such as remote sensing image segmentation, land cover classification, and phenotypic indicator monitoring. By leveraging these algorithms, it becomes possible to effectively and accurately monitor the growth of oil-seed crops by seamlessly integrating multi-source remote sensing and non-remote sensing data. In this study, we selected four commonly used machine learning regression prediction algorithms: PLSR, NLR, SVR, and BP-NN.

PLSR is an extension of the least squares method that effectively addresses the issue of multicollinearity among variables. It offers simplicity in computation and high predictive accuracy. Assuming the input data of the model, denoted $X$, are in the form of an $N \times M$ dimensional matrix, and the corresponding model output, denoted $Y$, is an $N \times 1$ dimensional matrix, performing matrix decomposition on the input and output yields the result shown in Equation (15):

$$\begin{aligned} X &= TP + B_1 \\ Y &= UQ + B_2 \end{aligned} \tag{15}$$

where $T$ and $U$ are the component score matrices of $X$ and $Y$, $P$ and $Q$ are the factor loading matrices, and $B_1$ and $B_2$ are the residuals fitted by the PLS algorithm. A regression relation $U = TE$ ($E$ is the regression coefficient matrix) is established for the component score matrices $T$ and $U$ of model input $X$ and output $Y$, and by substitution, Equation (16) can be obtained:

$$Y = ETQ + B_2 \tag{16}$$

A linear regression prediction model can be built when the response value $Y$ corresponds to the growth and physiological parameter indices of rapeseed seedlings and the input $X$ corresponds to multiple sources of data with different modalities.

NMR is an extension of multiple linear regression. Multiple linear regression establishes the relationship between two or more input variables $X$ and output variable $Y$. By combining the optimal combination of multiple interrelated factors of input variables, the output variable is jointly predicted or estimated. However, in practical applications, many of the input variables do not exhibit purely linear relationships with the output variables. By introducing interaction terms, non-linear variables can be combined to fit the non-linear part of the output variables. Assuming that $y$ is the output variable and $x_1, x_2, \ldots, x_k$ is the input variable, the model of NMR can be represented as shown in Equation (17):

$$y = \beta_0 + \sum_{i=1}^{k} \beta_i x_i + \sum_{i=1}^{k} \sum_{j=1}^{k} \beta_{ij} x_i x_j + \varepsilon \tag{17}$$

where $\beta_0$ is the constant term, $\beta_i$ is the linear regression coefficient, $\beta_{ij}$ is the nonlinear regression coefficient, and $\varepsilon$ is the fitting error.

SVR is a nonlinear regression method suitable for solving small-sample, high-dimensional problems. If the sample data used for regression training are $x_i$ and $y_i$, where $i = 1, 2, \ldots, n$, $x_i$ is the sample value of the input vector $x$ consisting of $n$ training patterns, and $y_i$ is the corresponding value of the desired model output. Then, the output of the regression model $y_i$ can be expressed as shown in Equation (18):

$$y\prime = w^T \phi(x) + b \tag{18}$$

where the coefficients $w$ and $b$ are adjustable model parameters, $w$ is a one-dimensional array, and $\phi(x)$ is a nonlinear transformation function that maps the input space to a high-dimensional feature space. The parameters $w$ and $b$ in the equation are then estimated by minimizing the cost function $J(w, \xi, \xi_i^*)$, which is defined by Equation (19):

$$\begin{aligned} \min \quad & J(w, \xi, \xi_i^*) = \tfrac{1}{2} \|w\|^2 + C \sum_{i=1}^{N} (\xi + \xi_i^*) \\ s.t. \quad & y_i - y_i\prime = \varepsilon + \xi_i, i = 1, 2, \ldots, N \\ & y_i\prime - y_i = \varepsilon + \xi_i^*, i = 1, 2, \ldots, N \\ & \xi_i \geq 0, \xi_i^* \geq 0, i = i = 1, 2, \ldots, N \end{aligned} \tag{19}$$

where $\xi_i$ and $\xi_i^*$ are positive relaxation variables, $\varepsilon$ is the distance between $y_i$ and $y_i\prime$, and $C$ is a positive real constant. The values of $w$ and $b$ in the equation and their modal outputs are obtained by calculating them in MATLAB software.

BPNN is an algorithm that uses error back propagation, which mainly consists of an implicit layer, an input layer, and an output layer. By back propagating the mean square error to the input layer, the connection weights between each neural layer are continuously modified until the actual output value has the minimum error with the predicted value.

## 4. Results and Discussion

### 4.1. Correlation Analysis

To avoid possible overfitting problems in monitoring models, researchers have explored the application of the Spearman correlation coefficient to perform multivariate correlation analysis on the independent variables (multi-source remote sensing data and multi-source non-remote sensing data) and dependent variables (physiological parameters of oil-seed rape growth) used in model construction [49]. The Spearman coefficient enables the analysis of the correlation between variables that do not conform to normality assumptions in the data. Figure 8 shows the test results of data normality, where the diagonal line of the matrix plot is a univariate density plot, which can show the type of data distribution of the variables. The other parts of the scatter matrix plots represent the linear

correlation between different variables. In this study, the correlation analysis of Spearman was chosen because the linear correlation between the multi-source data and the oil-seed rape growth physiological data was poor and only a small portion of the data satisfied the normal distribution.

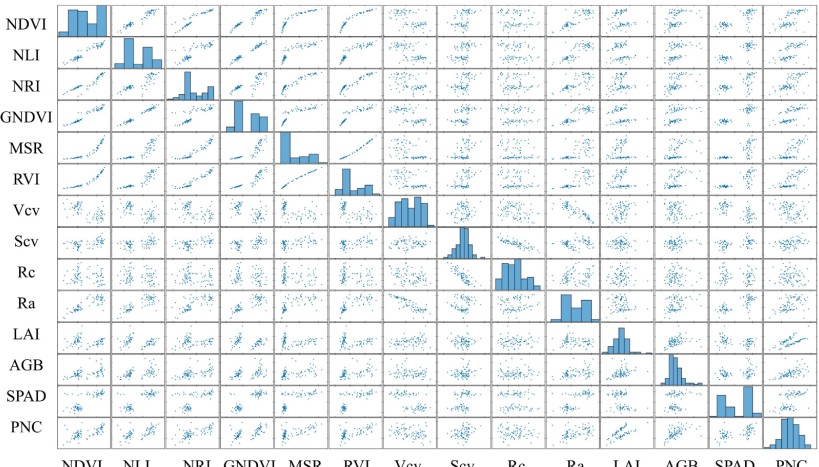

**Figure 8.** Scatter matrix of multi-source data variables of the model.

Figure 9 shows the Spearman correlation multivariate analysis; the correlation coefficients between different independent variables, and between independent variables and dependent variables, are shown in the figure, and the correlation coefficients are visualized in a heat map. Among these, the correlation between the vegetation closeness, $R_c$, of the multi-source remote sensing data and all four growth physiological indicators of oil-seed rape was poor, with correlation coefficients below 0.1. This may be related to the period of the crop, and the parameter is more suitable for modeling during the period from sowing to seedling emergence of oil-seed rape, rather than the period when the crop canopy covers a large area of soil. In addition, the correlation of the variables of the six spectral indices based on multi-source remote sensing data is high, which are prone to dimensional disasters when involving high-dimensional problems, leading to model overfitting; thus, it is necessary to reduce the dimensionality of the six spectral indices.

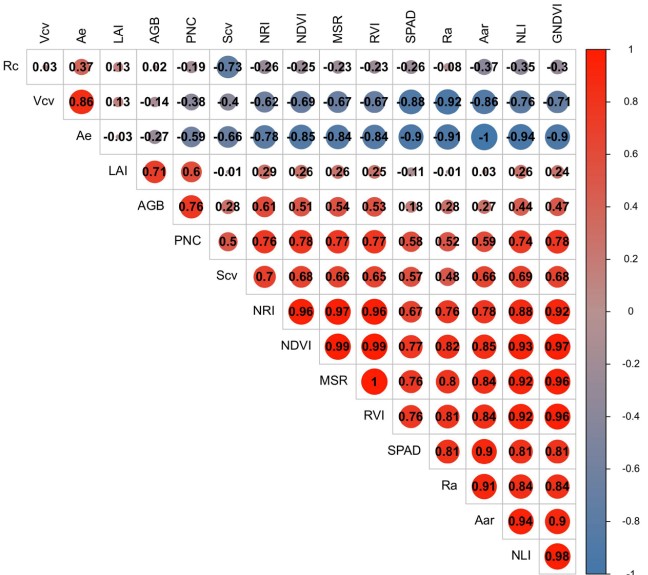

**Figure 9.** Spearman correlation analysis.

*4.2. PCA Data Dimensionality Reduction*

This study uses principal component analysis (PCA) to reduce the dimensionality of spectral indices. PCA, as an algebraic theory-based data dimensionality reduction method, can transform multiple variables into several linearly uncorrelated orthogonal vectors to completely represent the decision space [50,51]. There exists a certain degree of linear correlation among the original six spectral indices, allowing for the synthesis of information and features among multiple variables using a reduced set of composite variables. The raw data were processed using PCA to obtain the contributions of the variables, and variables were arranged in order from the largest to the smallest contribution. The combination of variables with a cumulative contribution of 95% was selected as the result of data dimensionality reduction. Figure 10 illustrates the distribution of variable contributions and the three-dimensional representation of the spectral data after dimensionality reduction. Among these, the cumulative contribution of NDVI, NRI, and MSR reached 95%; thus, the original spectral indices can be effectively reduced to this combination of three variables.

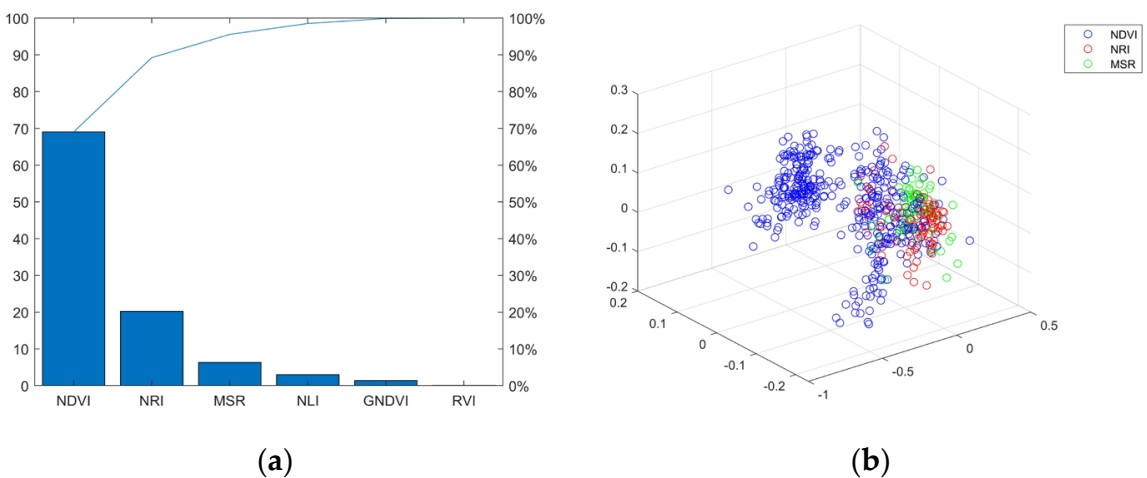

       (**a**)                    (**b**)

**Figure 10.** PCA dimensionality reduction result: (**a**) cumulative contribution of variables; (**b**) data distribution after dimensionality reduction.

In summary, after a series of data analyses and processing, the multi-source data and model outputs used for the monitoring model of oil-seed rape growth parameters in the field are: (1) multi-source remote sensing data after dimensionality reduction: spectral indices NDVI, NRI and MSR; texture features $V_{cv}$, $S_{cv}$ and $R_a$; (2) multi-source non-remote sensing data: meteorological data $A_e$ and $A_{ar}$; and (3) growth physiological parameters: LAI, AGB, SPAD, and LNC.

*4.3. Phenotypic Prediction of Rapeseed Crops during Seedling Stage*

4.3.1. Four Multimodal Data Model Frameworks

To ascertain the efficacy of utilizing diverse data sources for monitoring the growth of rapeseed during the early stages in agricultural fields, this study constructed a four-input Model Framework for Multimodal data (MFM) based on distinct multimodal data inputs: (1) a UAV remote sensing monitoring model based on a single data source; (2) a UAV remote sensing monitoring model based on a single data source (adding texture features); (3) a UAV remote sensing monitoring model based on the fusion of multi-source remote sensing data; and (4) a UAV remote sensing monitoring model based on the fusion of multi-source remote sensing data and multi-source non-remote sensing data.

Table 4 shows the differences between the four model frameworks. MFM1 used the field remote sensing monitoring method chosen by many studies, which only acquired MS images through a single data source (multispectral sensor) from UAV remote sensing, and, after processing, obtained spectral indices (VIs) to establish a growth monitoring

model of oil-seed rape in fields [52–54]. MFM2 added spatial texture feature information, which is a common method used to enhance the accuracy of monitoring models [55,56]. MFM3 is a remote sensing monitoring method based on the fusion of multi-source remote sensing data, in which MS images and RGB images were obtained from multiple data sources (multispectral sensors and visible sensors) of UAV remote sensing. Following the application of an image fusion algorithm, a high-resolution HR_MS image was obtained, from which spectral indices (VIs_F) and texture features (GCFs_F) were derived [57]. The final model, MFM4, is a large field monitoring model framework that integrated multi-source remote sensing data and multi-source non-remote sensing data. On the basis of MFM3, effective cumulative temperature $A_e$ and average rainfall $A_{ar}$ were added as model correction variables to enhance the interpretability and robustness of the model, and to improve the model's performance in different application scenarios.

**Table 4.** Four multimodal data model frameworks.

| Model Framework | Data Source | Model Input |
|---|---|---|
| MFM1 | Original MS images | Spectral index VIs (NDVI; NRI; MSR) |
| MFM2 | Original MS images | Spectral index VIs (NDVI; NRI; MSR) <br> Texture feature GCFs ($V_{cv}$; $S_{cv}$; $R_a$) |
| MFM3 | HR_MS image after image fusion | Spectral indices VIs_F (NDVI_F; NRI_F; MSR_F) <br> Texture Features GCFs_F ($V_{cv}$_F; $S_{cv}$_F; $R_a$_F) |
| MFM4 | HR_MS image after image fusion. Meteorological Data | Spectral indices VIs_F (NDVI_F; NRI_F; MSR_F) <br> Texture Features GCFs_F ($V_{cv}$_F; $S_{cv}$_F; $R_a$_F) <br> Meteorological data MDs ($A_e$; $A_{ar}$) |

### 4.3.2. Forecast Results

The study used the root mean square error (MSE) and the coefficient of determination, $R^2$, to evaluate the predictive effect of the model.

For machine learning modeling, 80% of the data were allocated as the training set, while the remaining 20% served as the test set. The holdout cross-validation method was used to divide the training and test sets, with the training set used to train the model parameters and the test set used to evaluate the accuracy and error of the model. Figures S1–S4 show the estimation results of the four modal inputs for the four growth physiological parameters of oil-seed rape, respectively. The plots show the true and predicted values of the test set in each model, as well as the coefficient of determination and mean square error of the evaluated models. Each plot represents the four model frameworks, MFM1, MFM2, MFM3, and MFM4, from the first row to the fourth row, and the first column to the fourth column represents the four machine learning models.

From Figures S1–S4, it can be seen that models with different frameworks have different accuracy in estimating the growth physiological indicators of rapeseed seedlings, indicating that the use of drone multispectral images combined with machine learning algorithms can indeed effectively estimate field crop phenotype information [14–16]. The monitoring model based on MFM4 has the highest accuracy and MFM1 has the lowest accuracy using only modal inputs from a single data source. The accuracy of the model frameworks for each type of modal input was ranked from largest to smallest as MFM4, MFM3, MFM2, and MFM1. There are several factors that lead to this situation. First, considering the addition of texture features and meteorological data, the increase in data type, such as the addition of meteorological data in MFM4 compared to MFM3, improves the practicability and generalization ability of the model. Second, due to multi-source remote sensing image fusion, the quality of remote sensing images is enhanced while the experience quality (QoE) is increased. The accuracy of the model established using high-resolution remote sensing images is improved and the root mean square deviation is reduced [21]. The results show that multi-source data can effectively enrich the complementary information from different sensors. The superposition model, MFM3, which belongs to the multi-modal data based on the image fusion algorithm, shows an advantage over

MFM2 in terms of model accuracy and adaptability. This is in line with the conclusions of previous studies, and is important for establishing a reliable physiological monitoring model for oilseed rape growth [18,24,25].

In this study, a model for quantitative remote sensing monitoring of winter oilseed rape seedling growth status based on multimodal data is proposed, and four modeling frameworks with different inputs are compared. Quantitative analysis of the differences between the different modal model frameworks is the focus of this study. The results of the machine learning model evaluation based on the four multimodal model frameworks are shown in Table 5; the best values of mean square error and coefficient of determination for the monitoring models built with the four sets of modal inputs are highlighted to represent the best model for that modal input [58,59]. The best model accuracy and mean square error built with MFM1 were $R^2$ = 0.5730 and MSE = 7.7398, while those with MFM2 were $R^2$ = 0.6350 and MSE = 7.5148. The difference between the inputs of the two modalities was that the latter had an additional set of texture features, which is an improved method used by many agricultural remote sensing researchers, and the addition of texture features had a facilitative effect on building a more accurate model. The best model accuracy and mean square error of MFM3 was $R^2$ = 0.7183 and MSE = 7.3309, which was a 13.1% improvement in prediction model accuracy and a 2.4% reduction in mean square error compared to MFM2. MFM3 was based on MFM2 to make changes to the source MS images used to extract spectral indices and texture features. Based on the RMGF image fusion algorithm used in this study, the high-resolution HR_MS images were obtained by making full use of the spectral features of MS and the spatial structure features of RGB; the complementary information of multi-source remote sensing data from two sensors was combined to reduce the inhibiting effect of a single information source. This formed a complete and consistent information description of the target. It is thus possible to draw the conclusion, consistent with the previous study, that feature fusion can solve problems such as data redundancy and multicollinearity, thus improving the accuracy of the model [60]. The best model accuracy and mean square error of MFM4 was $R^2$ = 0.7454 and MSE = 6.6630, which improved the model accuracy by 3.7% and reduced the mean square error by 9.1% compared to MFM3. The modal input added multi-source non-remote sensing data, i.e., meteorological data obtained from different sensors, and the improvement in accuracy was smaller, but the mean square error of the model was significantly reduced. This proved that the addition of multi-source non-remote sensing data can improve the robustness of the model to enhance its ability to be generalized to different application scenarios and ensure its modeling effectiveness in other scenarios or scales [31,61].

The modeling effects of different machine learning models varied under different modal inputs, as well as for different physiological indicators of oil-seed rape growth. In Table 5, the BPNN algorithm obtained the highest coefficient of determination and the NMR algorithm obtained the lowest mean square error for the modal input of MFM1; the BPNN obtained the best model parameter values for the modal input of MFM2; and the SVR obtained the best model parameter values for both MFM3 and MFM4. When the data sources were small, NMR showed the ability to provide a solution in low-dimensional space; however, when the data sources were large, NMR had too many linear and nonlinear terms, which may generate singular matrices during the resolution process and cause overfitting of the model [62]. The PLSR did not obtain the best model parameters, mainly because PLSR is a fitting process based on the component contributions, which may be better in solving the problem of multiple covariance [63]. However, in this study, the data with high correlation were dimensionally reduced before modeling, and the data had weak multicollinearity and the modeling effect of PLSR was poor. The performance of the models also varied for different physiological indicators of seedling rape growth. For instance, the SVR algorithm consistently achieved the highest accuracy in the prediction of LAI, while NMR obtained the majority of optimal estimates in the prediction of AGB.

**Table 5.** Machine learning models for four multimodal model frameworks to evaluate physiological indicators of rapeseed growth.

| Modal Input | Machine Learning Models | Evaluation Indicators | Physiological Indicators of Oil-Seed Rape Growth | | | | Average |
|---|---|---|---|---|---|---|---|
| | | | LAI | AGB | LNC | SPAD | |
| MFM1 | SVR | R-square | 0.6773 | 0.3864 | 0.4729 | 0.7237 | 0.5651 |
| | | MSE | 0.2068 | 32.1681 | 6.0256 | 15.2862 | 13.4217 |
| | PLSR | R-square | 0.4528 | 0.3391 | 0.1971 | 0.2183 | 0.3018 |
| | | MSE | 0.4042 | 28.7874 | 11.6831 | 41.9419 | 20.7042 |
| | BPNN | R-square | 0.4233 | 0.3215 | 0.5304 | 0.8277 | 0.5257 |
| | | MSE | 0.3649 | 15.2748 | 4.1654 | 9.1541 | **7.7398** |
| | NMR | R-square | 0.3654 | 0.4482 | 0.7043 | 0.7741 | **0.5730** |
| | | MSE | 0.4262 | 16.5012 | 3.8428 | 12.3704 | 8.2852 |
| MFM2 | SVR | R-square | 0.7029 | 0.4697 | 0.4829 | 0.7784 | 0.6085 |
| | | MSE | 0.2249 | 20.8486 | 7.2676 | 12.0857 | 10.1067 |
| | PLSR | R-square | 0.5212 | 0.3406 | 0.2846 | 0.4586 | 0.4013 |
| | | MSE | 0.3601 | 39.9055 | 7.8961 | 29.3944 | 19.3890 |
| | BPNN | R-square | 0.6253 | 0.4477 | 0.57875 | 0.8881 | **0.6350** |
| | | MSE | 0.2641 | 17.9229 | 4.996 | 6.8762 | **7.5148** |
| | NMR | R-square | 0.4281 | 0.5019 | 0.6239 | 0.8079 | 0.5905 |
| | | MSE | 0.3731 | 38.6885 | 5.7252 | 8.0058 | 13.1982 |
| MFM3 | SVR | R-square | 0.7802 | 0.5909 | 0.6371 | 0.8651 | **0.7183** |
| | | MSE | 0.1471 | 17.8331 | 3.7256 | 7.6178 | **7.3309** |
| | PLSR | R-square | 0.6298 | 0.4514 | 0.4145 | 0.5975 | 0.5233 |
| | | MSE | 0.2919 | 29.2122 | 5.8248 | 19.7959 | 13.7812 |
| | BPNN | R-square | 0.6505 | 0.4703 | 0.5991 | 0.8935 | 0.6534 |
| | | MSE | 0.2962 | 27.1946 | 4.4556 | 5.2076 | 9.2885 |
| | NMR | R-square | 0.4466 | 0.5918 | 0.7027 | 0.8202 | 0.6403 |
| | | MSE | 0.3853 | 17.6365 | 4.129 | 9.4111 | 7.8155 |
| MFM4 | SVR | R-square | 0.8071 | 0.6356 | 0.6646 | 0.8742 | **0.7454 *** |
| | | MSE | 0.1411 | 17.4372 | 3.3715 | 5.5718 | **6.6630 *** |
| | PLSR | R-square | 0.5973 | 0.4903 | 0.4494 | 0.6526 | 0.5474 |
| | | MSE | 0.3251 | 18.7233 | 5.7181 | 15.7756 | 10.1355 |
| | BPNN | R-square | 0.7702 | 0.4438 | 0.6351 | 0.8852 | 0.6836 |
| | | MSE | 0.1222 | 33.0606 | 6.1542 | 5.6177 | 11.2387 |
| | NMR | R-square | 0.6045 | 0.5602 | 0.6915 | 0.8266 | 0.6707 |
| | | MSE | 0.2539 | 24.1886 | 3.5773 | 8.5108 | 9.1327 |

Note: * represents the best results of this model framework for estimating physiological parameters of seedling oil-seed rape growth. The bold represents the best value of mean squared error and coefficient of determination, and also represents the best model of the modal input.

## 5. Conclusions

This study focused on the early-stage growth of oil-seed rape and utilized data from unmanned aerial vehicle (UAV) visible-light images, multispectral images, and four crucial growth physiological indicators measured in real time. Four MFMs with different modal inputs were proposed for physiological monitoring of rapeseed growth during the early stage, and four machine learning models, SVR, PLS, BPNN, and NMR, were used to compare the differences between multi-source data and single-source data in monitoring of oil-seed rape using remote sensing. The results demonstrate that the models, which incorporate VIs and GCFs extracted from an image fusion algorithm, as well as effective accumulated temperature ($A_e$) and average rainfall ($A_{ar}$) as correction variables, effectively leveraged complementary information from various UAV remote sensing data sources. This mitigated the inhibitory effect of single-source data on the models, which were able to accurately detect rapeseed growth during the early stage. Among these, the SVR model based on multi-source remote sensing data and multi-source non-remote sensing data exhibited high accuracy with minimal error, showcasing robustness and generalizability in various scenarios. This research provides a theoretical basis for precise field management and agricultural production.

**Supplementary Materials:** The following supporting information can be downloaded at: https://www.mdpi.com/article/10.3390/rs15163951/s1, Figure S1: Prediction results of LAI for four multimodal models; Figure S2: Prediction results of AGB for four multimodal models; Figure S3: Prediction results of LNC for four multimodal models; Figure S4: Prediction results of SPAD for four multimodal models.

**Author Contributions:** Conceptualization, G.Z., Y.Y. and Y.R.; methodology, J.Z.; software, J.W. (Jiang Wang); validation, J.W. (Jiang Wang); formal analysis, X.W.; investigation, Y.Y.; resources, J.W. (Jian Wang) and Z.J.; data curation, J.Z.; writing—original draft preparation, X.W.; writing—review and editing, X.W.; visualization, J.W. (Jiang Wang); supervision, G.Z. and Y.Y.; project administration, G.Z. and Y.Y. All authors have read and agreed to the published version of the manuscript.

**Funding:** This research received no external funding.

**Conflicts of Interest:** The authors declare no conflict of interest.

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
