# Peer review of "Prediction of Seedling Oilseed Rape Crop Phenotype by Drone-Derived Multimodal Data"

_remotesensing, doi:10.3390/rs15163951_

Round 1
Reviewer 1 Report
This paper presents a seedling oilseed rape crop phenotype prediction method for from drone-derived multimodal data.
The studied topic is meaningful.
The paper still has some major problems.
The authors are suggested to revise the paper given the following comments.
As discussed in some surveys and studies (Perceptual image quality assessment: a survey; Screen content quality assessment: overview, benchmark, and beyond; Blind quality assessment based on pseudo-reference image; Blind image quality estimation via distortion aggravation; Unified blind quality assessment of compressed natural, graphic, and screen content images), quality of experience (QoE) is an important aspect of various intelligent systems. High-quality images/videos are important for the successful usage of these intelligent systems, while low-quality media may degrade the performance of these systems.
The authors may give some discussions on this aspect and the above mentioned works.
Hazy scenes are frequently observed in remote sensing images, and the haze will degrade the performance of the following analysis, as discussed in ‘Objective quality evaluation of dehazed images’ and ‘Quality evaluation of image dehazing methods using synthetic hazy images’. The authors may give some discussions on this aspect and the above mentioned works.
The authors integrate information of multiple modalities for analysis. More multimodal studies are suggested to be discussed. As described in some works (Study of subjective and objective quality assessment of audio-visual signals; A multimodal saliency model for videos with high audio-visual correspondence; Fixation prediction through multimodal analysis), fusing audio-visual information can improve some single modality visual analysis systems.
The authors may give some discussions on this aspect and the above mentioned works.
The writing of the paper still needs improvement.
None
Reviewer 2 Report
This manuscript (remotesensing-2520274) demonstrated using Unmanned Aerial Vehicle (UAV) remote sensing and Image Guided Filtering Fusion (GFF) to evaluate the growth of rapeseed seedlings through key indicators like Leaf Area Index, Above Ground Biomass, Leaf Nitrogen Content, and Chlorophyll Content. Utilizing various machine learning models, the study found that integrating both remote and nonremote sensing data, including temperature and precipitation, achieved the highest precision, with support vector regression consistently performing well across different frameworks, offering a valuable reference for nondestructive crop monitoring.
The introduction, objectives, materials and methods, results and discussion (in a single topic), and conclusions are well-written and appropriate for the manuscript. The figures and tables are also very informative; I have only minor reservations for the authors, which I will detail in the following comments.
Keywords in alphabetic order;
L20. Scientific name in italics. In addition, check in all manuscript.
Regarding the figure captions, consider making them as precise as possible in their description. For example, include all abbreviations and measures that were presented, the sample size, a detailed description of what it refers to, and the plant used (as in the scheme of Figure 1). In Figure 5, what does "Acquisition process of gradient feature data" refer to? Additionally, in the equations, refer to them as "Eq." or "Equation," and properly reference them in the text.
Figure 10 requires corrections. It appears cut off, so I cannot see the cumulative frequency of the PCA or the explanation values of PC1, PC2, PC3. Please consider adding a legend for the colors. What do they represent?
All abbreviations appearing for the first time must be fully spelled out, especially in the materials and methods and results and discussion sections. Please kindly check this throughout the manuscript.
L429-452. From the text, it is not clear whether this is just the results section or includes discussion as well? There are instances of discussion but no references. Could you please add appropriate references?
I believe that Figures 11, 12, 13, and 14 should be moved to the supplementary materials, as they do not provide essential information for understanding the text. Alternatively, the authors could consider revising and combining them.
Lines 477-515; and; 520-536 need additional references. The text includes discussion but lacks theoretical and referential grounding to justify the results found.
Table 5: What do the bold values and *(asterisk) represent?
Check old references and change for more recently.
Reference 19. Translator or English;
Minor English language editing is required for grammar, spelling, and some redundancy found, especially in sentences that do not contain citations/references.
Reviewer 3 Report
1. in section 4.3.1, in the four model frameworks mentioned. MFM3 and MFM4 have higher spatial resolution than MFM1 and MFM2. How did you compare the model prediction accuracies estimated from different spatial resolution MFM? Did you do any post-processing like data binning to force the predicted results have the same spatial resolution?
2. In section 2.3, could you please explain more about the NIR band? For example, why 730nm is selected? What is the bandwidth of NIR band? This is crucial because NDVI is very sensitive to the spectral wavelength used to calculate it, as reported in "Potential of red edge spectral bands in future landsat satellites on agroecosystem canopy green leaf area index retrieval". Please cite this paper and add more details about central wavelength used to calculate NDVI.
3. in section 2.4.1, which signal did you use for this study? Radiance signal or calibrated reflectance signal? Please explain.
4. in section 2.3, did you consider the collected radiance signal non-uniformity due to detector relative spectral response (RSR) shift effect as reported in "Impact of wavelength shift in relative spectral response at high angles of incidence in landsat-8 operational land imager and future landsat design concepts"? Since the instrument DFOV is 56x84 degree, which will introduce radiance signal variance and result in error. Please cite this paper and clarify if RSR shift impact is considered.
5. in section 4.3.2, no need to explain the details of RMSE and R2, please remove Equation (20) and (21)
Round 2
Reviewer 1 Report
Most of the previous concerns are addressed. However, all previously recommended topics and studies are suggested to be discussed in the paper. Moreover, the formats of all references are suggested to be double-checked.
None
Reviewer 3 Report
n/a
Author Response
Thank you very much for your valuable suggestion and correction.